# Al-Cu-Mg Alloy Powder Reinforced with Graphene Nanoplatelets: Morphology, Flowability and Discrete Element Simulation

Mulla Ahmet Pekok [1], Rossitza Setchi [1,*], Michael Ryan [1], Heng Gu [2], Quanquan Han [3] and Dongdong Gu [4]

1   Cardiff School of Engineering, Cardiff University, Cardiff CF24 3AA, UK
2   School of Mechanical Engineering, Jiangsu University, Zhenjiang 212013, China
3   School of Mechanical Engineering, Shandong University, Jinan 250061, China
4   College of Materials Science and Technology, Nanjing University of Aeronautics and Astronautics, Nanjing 210016, China
*   Correspondence: setchi@cardiff.ac.uk; Tel.: +44-(0)29-2087-5720

**Abstract:** Research in metal matrix composites (MMCs) indicates that superior mechanical properties may be achieved by embedding reinforcement materials. However, the development of new composite powder for additive manufacturing requires an in-depth understanding of its key characteristics prior to its use in the fabrication process. This paper focuses on the low-energy ball milling (LEBM) of aluminium 2024 alloy (AA2024) reinforced with graphene nanoplatelets (GNPs). The main aim is to investigate the effect of the milling time (from 0.5 to 16 h) on the morphology and flowability of the powder. The study shows that, while short milling times (under 2 h) could not break the Van der WaRals forces between nanoparticles, GNPs were well separated and sufficiently covered the powder surface after 4 h of milling, thanks to the continuously applied impact energy. Longer milling time provides increasingly similar flowability results, confirmed by both the experimental work and discrete element model (DEM) simulations. Moreover, the ball milling process decreases the crystallite size of the milled powder by 24%, leading to a 3% higher microhardness. Lastly, the surface energy of the powder was determined as 1.4 mJ/m$^2$ by DEM, using the angle of repose of the as-received powder from experimental work.

**Keywords:** ball milling parameters; graphene nanoplatelets; aluminium 2024 alloy; discrete element model; powder characterization

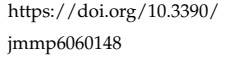



## 1. Introduction

Many engineering sectors (i.e., aerospace, automotive, marine, construction, and defence) require materials with superior characteristics such as damage tolerance, strength, fatigue resistance, corrosion resistance, low density, conductivity, low cost, and recyclability for their applications [1]. Although aluminium (Al) and its alloys satisfy their demands in terms of damage tolerance, corrosion resistance, and density, their poor mechanical properties, such as low strength, need to be improved. Creating metal matrix composites (MMCs), by adding materials to the metal for reinforcement, is one method of improving the mechanical properties of Al alloys. The use of graphene (Gr) as a reinforcement material has gained much attention owing to its superior properties, such as high elastic modulus (1 TPa), high mechanical strength (130 GPa), and excellent thermal (5300 W/mK) and electrical (6000 S/cm) conductivities [2,3]. However, despite these highly advantageous properties, using Gr as a reinforcement material has its own unique challenges, particularly when producing powder for additive manufacturing (AM) processes such as laser powder bed fusion (LPBF), where the final component properties are sensitive to the characteristics of the powder [4]. Phenomenon which can affect the behavior of the powder, such as

powder flattening and agglomeration, are commonly observed during the milling stage due to inappropriately selected milling parameters, strong interlayer Van der Waals forces, the density difference between the alloy (2.7 g/cm$^3$) and Gr (1.8–2.2 g/cm$^3$) and massive surface area of Gr [5–7].

Ball milling (also known as mechanical milling or mechanical alloying) is a dry powder processing technique for producing small quantities of powder. The process involves cold-welding, fracturing, diffusion, and deformation of particles, owing to the repeatedly applied impact energy [8,9]. Ball milling offers a wide range of parameters (such as milling and pause times, milling speed, process control agent (PCA), weight ratio, different sizes of milling balls and different milling atmospheres) which can be selected in order to achieve the desired powder morphology and reinforcement particle distribution [10]. High impact and shear forces (generated at faster milling speeds and longer milling times) can break the interlayer Van der Waals forces between the Gr sheets and eliminate the agglomeration [5]; however, applying higher impact energy than is adequate to break these strong bonds comes with an equally significant challenge prior to the LPBF process, namely flat powder morphology. Thus, metal and ceramic powders that have been homogeneously mixed by ball milling may be mechanically alloyed under high energy until a steady state is obtained [11]. Powder morphology can significantly affect the AM process and should therefore be carefully considered in order to achieve adequate flowability of the milled powder [12]. Poor-flowing powders create non-uniform layers and low powder bed density, which dramatically affects the final part quality [13,14]. Additionally, another known gap is the effect of Gr concentration in composites on powder flowability. Therefore, it is important to identify appropriate ball milling parameters which sufficiently reduce the agglomeration of Gr and distribute it evenly in the Al matrix, while retaining a powder morphology that is suitable for processing via LPBF.

Studies of Gr-reinforced Al and its alloys substantiate that Gr is capable of increasing some mechanical properties of the fabricated composites. For instance, pure Al and different weight ratios of multi-layer Gr sheets (0.5, 1 and 2.5 wt.%) were milled and samples fabricated using LPBF, and it is reported that the microhardness was improved by 75%, with the addition of Gr (2.5 wt.%) [15], which suggests a uniform distribution of Gr within the matrix and its efficient involvement in grain refinement [16]. It was also reported that the existence of Gr, grain refinement, the influence of thermal mismatch and the Orowan strengthening mechanism are responsible for the improvement [17]. Another study, examining a few layers of graphene (FLG)-reinforced AA2024, produced using high-energy ball milling (HEBM) and hot rolling stated that the yield strength (YS) of the product improved up to 100% with the addition of 0.7 wt.% FLG [18]. However, a wholly flattened powder morphology (which negatively affects the flowability of the composite) was determined following milling in this study [18].

Previous studies have explored powder particles under different milling speeds. For example, HEBM and low-energy ball milling (LEBM) applied to AA2024 powder reinforced with graphene nanoplatelets (GNPs) have been employed to study morphological evolution [19]. It was reported that higher impact energy inside the bowl (created by HEBM) immediately changes the powder morphology to flat, which negatively affects the powder flowability, and Gr is embedded into the matrix powder [19]. The LEBM, however, gradually changed the powder morphology of the composite powder to flat, and dispersed GNPs start to adhere onto the powder surface at longer milling times [19]. Another significant factor of plastic deformation at this stage is the softness of metal powder. Under high impact energy, or continuously applied low impact energy, soft AA2024 powders instantly or eventually turn powder morphology flat. Moreover, the LEBM allows adequate time to observe the transition of powder morphology and Gr dispersion with milling time.

Building on the contributions of an aforementioned study [19], this paper further aims to explore the characteristics of the milled GNPs-reinforced AA2024 powder under a wide range of milling times (from 0.5 to 16 h) in terms of powder morphology, average crystallite size and compressibility of the composite powders. Moreover, the flowability of powder

with different percentages of Gr (0.1, 0.2 and 0.5%) is examined for the produced powder at various milling times. To understand the experimentally observed results, a discrete element model (DEM) model was performed with commonly used particles of literature and realistic DEM particles that are obtained from scanning electron microscope (SEM) images. Furthermore, the effect of single and multiple (3, 6, and 10) particles and different volumes of the multiple particles in the powder pool is studied, which is impossible to examine through experimental work. Exploring the characteristics of milled powder is essential to understand the suitability of the powder for the LPBF process. Hence, this study offers some practical suggestions for successfully preparing powder for use in LPBF.

## 2. Materials and Methods

### 2.1. Powder Specification and Characterisation

This study used a commercial gas atomized AA2024 powder ($-325$ mesh) from Carpenter Additive Technology Corporation (Philadelphia, PA, USA). Gas atomization is currently the most common commercial manufacturing process for Al and its alloys, as the high solidification rate of gas atomization produces powders with good homogeneity and a fine structure [20]. The composition of the alloy powder was 4.9Cu-1.8Mg-0.9Mn-0.5Si-0.5Fe-bal.Al (wt.%) [21]. This alloy may also contain small amount of Zn (<0.25%), Ti (<0.15) and Cr (<0.1) [22]. GNPs (15 μm particle size, 50–80 m$^2$/g surface area) were obtained from Sigma-Aldrich Company Ltd. (Dorset, UK). The Gr particles normally accumulate and create thick and large particles, as shown in Figure 1c. A Malvern Mastersizer-3000 (Malvern, UK) was used to determine the particle size distribution (PSD) of the as-received alloy and milled composites using the laser diffraction measurement method. In this technique, a laser beam passes through a particle sample, and a machine measures the angular variation in intensity of the scattered light. Then, using the Mie theory of light scattering, the angular scattering intensity data is analyzed to calculate the size of the reflected particles. Furthermore, particle size and volume density of $Dv_{10}$, $Dv_{50}$ and $Dv_{90}$ were obtained using the Malvern Mastersizer-3000 for as-received and milled powders. The median particle size ($Dv_{50}$) of the AA2024 was measured as 37.6 μm (see Figure 1). Accurate measurement of Gr particles using PSD technique is difficult because of the accumulation of the GNPs. Even though ultrasound wave has been used to separate the particles, measured Gr particles are larger than 15 μm particle size.

X-ray differentiation (XRD) and energy dispersive X-ray spectroscopy (EDS) analysis were conducted to determine the phases formed during the ball milling process. A Siemens/Bruker D5000 X-ray powder diffraction machine was used with Cu K$_\alpha$ radiation ($\lambda$ = 0.15406 nm) at 40 kV and 30 mA settings. The start and stop angles were selected as 15° and 90° with a step size (°2θ) of 0.02. The average crystallite size ($D$) was estimated as follows [23]:

$$D = (K \times \lambda) / (\beta \times \cos \theta) \tag{1}$$

where $K$ is a Scherrer constant close to unity (0.9), $\beta$ is the line broadening at full width at half maximum (FWHM) and θ is the Bragg's angle.

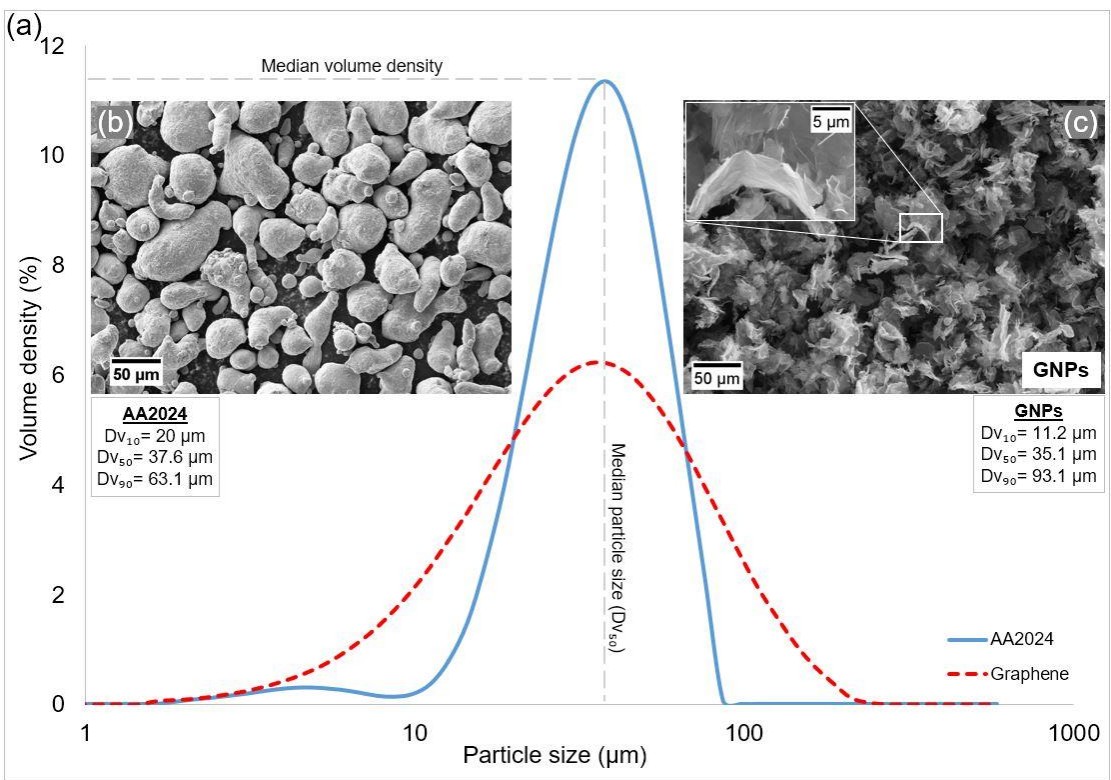

**Figure 1.** (**a**) The PSD of the alloy and Gr, (**b**) SEM images of the as-received (0 h) powder and (**c**) GNPs.

### 2.2. Powder Preparation

A laboratory planetary ball milling machine (PULVERISETTE 5 classic line, Fritsch GmbH, Idar-Oberstein, Germany) was employed to mill the alloy with Gr under different milling times, ranging from 0 to 16 h. The 0 h (zero hours) milling is representative of as-received powder without milling condition. Two milling bowls were placed opposite each other inside the milling machine, in order to balance the centrifugal force. Each bowl was filled with 800 g milling balls and 80 g powder (included 77 g AA2024, 1.6 g stearic acid ($C_{18}H_{36}O_2$), and 0.8 g GNPs). The ball-to-powder weight ratio was adjusted as 10:1 during the experiment. The milling bowls were loaded with approximately 40 stainless steel balls with two different diameters (10 and 20 mm) and weights (4 and 32 g) in order to vary and randomize the impact energy. It has also been reported that using a mix of big and small size balls while milling reduces the amount of cold welding and powder coating on the balls' surfaces [24]. Even though no precise reason for the enhanced yield under these conditions has been provided, it is plausible that the varied-sized balls cause shearing forces that aid in the detachment of the powder from the balls' surface [25]. Stearic acid (ranging from 1 to 3 wt.%) is a commonly used PCA for Al powders in literature and is reported as adequate to prevent the powder from contamination [26]. With the aid of a previous study [19], 2 wt.% stearic acid was used to create a thin film on milling balls as a PCA.

Milling speed was kept constant at 100 rpm, in light of the previous study [19], in order to eliminate the immediate changing of the powder morphology to flat. Seven different milling times from 0.5 to 16 h (see Table 1) have been used in experiments, in order to determine the effect of milling time and therefore establish the optimum milling time for the composite. Additionally, 10 min of milling followed by 10 min of pause time was used in order to prevent the powder from reaching high temperatures during the milling process (pause time is not included in the milling times stated).

**Table 1.** The processing parameters for the seven samples of the advanced Gr/AA2024 composites.

| Parameters | Values |
| --- | --- |
| Milling speed (rpm) | 100 |
| Total milling time (h) | 0.5/1/2/4/8/12/16 |
| Milling/pause time (min) | 10/10 |
| Ball-to-powder weight ratio | 10:1 |

*2.3. Flow Characterisation*

Universally, the flow characteristic of a powder is measured by detecting the angle of repose of the deposited powder using a powder flowability measurement kit, with the powder flowing through a funnel onto a stage. The angle of repose can be explained as the angle which differentiates the transitions among the phases of the granular materials, and it is directly related to the resistance to movement, or inter-particulate friction, between particles [27,28], and is measured as the angle between the heaped cone of a free-standing powder and the horizontal plane (hillsides) [28].

DEM is a numerical technique for simulating the dynamic behavior of a powder and can also be used to calculate the angle of repose in relation to Newton's law of motion [29,30]. The interaction of powder-to-powder and powder-to-wall helps to understand powder's flowability prior to use in the AM process.

Flowability tests were conducted using both DEM simulation and experimental work. The powder deposition model was designed using EDEM-5.0.0 (2019) simulation software. The model includes a powder tank, funnel, and floor (see Figure 2a). The funnel was designed according to the ASTM-B213-13 standard. In order to estimate the angle of repose of the raw and milled powder, three different powder particle templates (see Figure 2c) from the SEM images of real powder particles (see Figure 2b) for each milling time were created using computer-aided design software (SolidWorks 2019). GNPs were not included in the DEM simulation, as it is not practical to model a nano-size thin layer with a large surface area (as found in Gr flakes) with spherical DEM cells without requiring enormous working time for each simulation. The simulation analyzes the flowability of the milled AA2024 powder without Gr. Hereby, the difference between experimental work (with Gr) and simulation (without Gr) will allow the effect of Gr on the flowability of the milled powders to be examined.

Modelling particles at or below nano-scale is a significant challenge, owing to the limitations of computing power [31]. The simulations were produced using a computer of the following specification: Intel® Core™ i7-8665U CPU@1.90GHz 2.11 GHz, 16.0 GB Ram and 64-bit operating system. However, considerable computational time and high-performance computing may be required for some applications in order to simulate the size of the particles at the physical scale. A scale-up of the powder size is commonly preferred to alleviate this computational burden [31–35]. Furthermore, full-scale modelling is impractical for DEM-based simulations when complicated powder morphologies (such as non-spherical AA2024 powders) are considered. Additionally, it has been stated that upscaling has no substantial effect on the results, and some coefficient of frictions obtained from unscaled particles can be used for upscaled particles in order to simulate the angle of repose precisely [32]. Furthermore, the contact sliding and rolling coefficients of friction for various particle sizes were investigated and it was reported that the angle of repose was independent of the particle diameter [32]. Therefore, micron-sized powders in the present study were scaled-up in order to reduce the excessive running time of a large number of simulations.

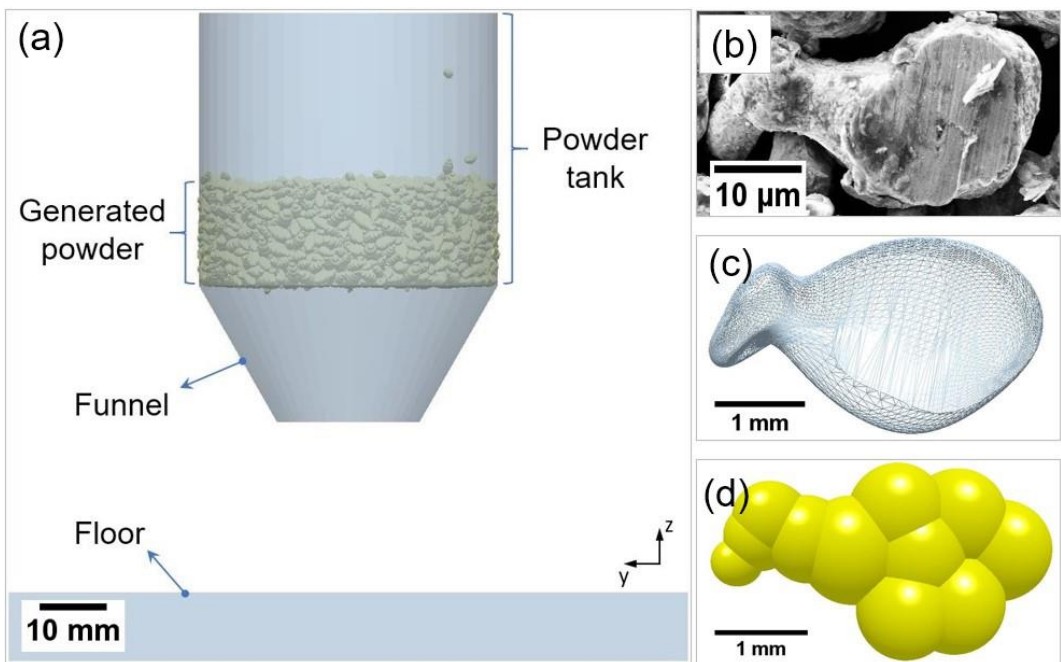

**Figure 2.** Projection of (**a**) simulation model with an example of (**b**) SEM image of the 0.5 h milled powder, (**c**) created a 3D template based on the SEM images of the real powder and (**d**) created DEM particle with 11 spherical cells from the template.

Hertz–Mindlin with the Johnson–Kendall–Roberts (JKR) is a cohesive model which is also accounting the influence of Van der Waals forces [36]. According to the JKR model, the normal elastic contact force ($F_{JKR}$) is expressed with the interfacial surface energy (Γ) as follows [36]:

$$F_{JKR} = -4\sqrt{\pi\Gamma E^*}a^{\frac{3}{2}} + \frac{4E^*}{3R^*}a^3 \qquad (2)$$

$$\Gamma = \gamma_1 + \gamma_2 - \gamma_{1,2} \qquad (3)$$

where $E^*$, $R^*$ and $a$ are the equivalent Young's modulus, radius, and contact radius and $\gamma_1$, $\gamma_2$ and $\gamma_{1,2}$ are the surface energy of two spheres and interface surface energy, respectively. $E^*$, $R^*$ can be defined as follows [37]:

$$\frac{1}{E^*} = \frac{1 - v_i^2}{E_i} + \frac{1 - v_j^2}{E_j} \qquad (4)$$

$$\frac{1}{R^*} = \frac{1}{R_i} + \frac{1}{R_j} \qquad (5)$$

where $E_i$, $R_i$, $v_i$ and $E_j$, $R_j$, $v_j$ are Young's modulus, radius, and Poisson's ratio of each spherical in contact, respectively.

Single spherical particle (Type 1a) and different combinations of several spherical particles (Type 1b–f), which are commonly used in literature, were tested in order to contrast the common particles (Type 1) and SEM particles (Type 2) (see Figure 3). In the beginning, combinations of different percentages (20, 30, 50, and 100%) of three particles (Type 2a–c) selected from SEM photos of as-received powder were compared in order to exhibit the effect of various percentages on the angle of repose. Additionally, three (Type 2a–c), six (Type 2a–f), and ten (Type 2a–j) different morphologies with equal percentages were analyzed in order to determine the optimum number of different particle morphologies.

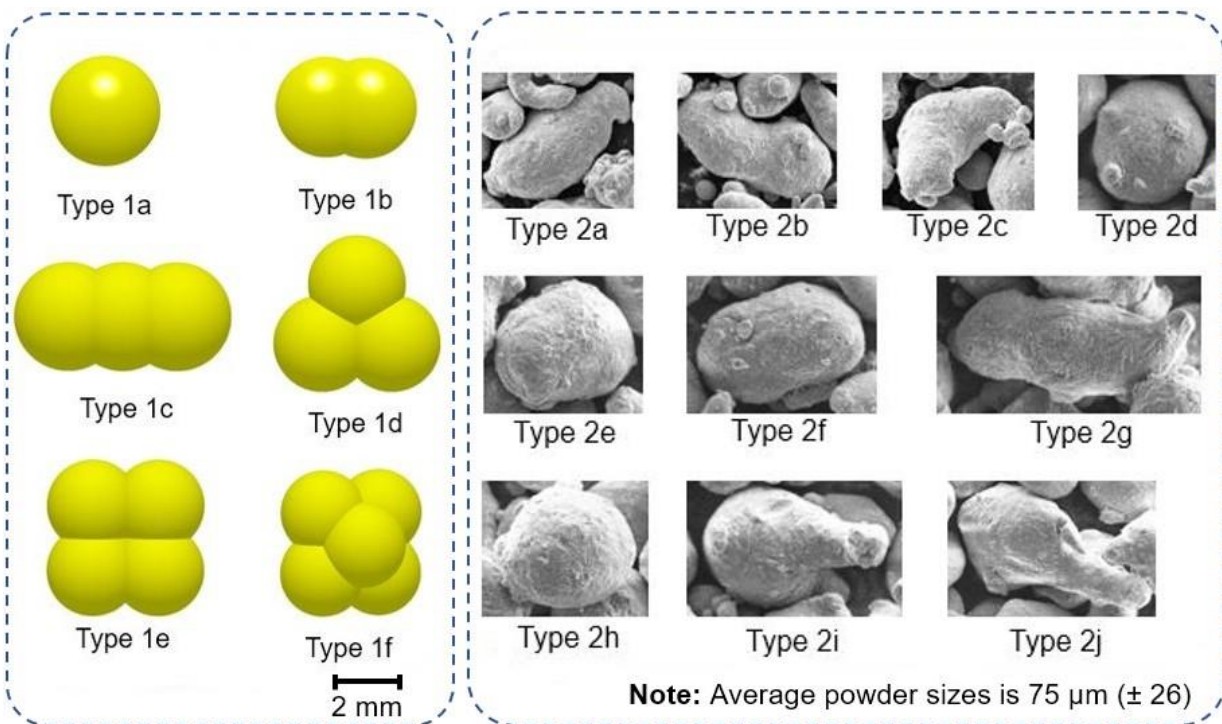

**Figure 3.** Commonly used particle morphologies in literature (Type 1 series) and the most representative particles for SEM photos of the real powders (Type 2 series).

The three most representative particles for as-received powder at each milling time were selected based on visual observation of SEM images to create templates. Examples of these images are represented in the following section (see as an illustration. Based on the templates, DEM particles were formed using a combination of 5 to 42 spherical cells (see Figure 4). Fewer spherical cells (5 to 14) are required for the alloys milled for up to 2 h, due to the dominant "round" particle shapes; however, a large number of cells (up to 42) were used to create milled particles over 4 h, because of the thin, flat particle shapes (see Figure 4). The PSD results from the experimental work were used to generate the powder's size in the powder pool of the simulations (see Figure 1).

The used DEM parameters are shown in Table 2. The particle surface energy parameter for the as-received powder was optimized using the angle of repose of the as-received powder in experimental work.

**Table 2.** The DEM parameters used in the flowability simulations.

| Parameters | Value | Ref. |
|---|---|---|
| Poisson's ratio | 0.33 | [38] |
| Solid density | 2768 kg/m$^3$ | [39] |
| Young's modulus | 73.08 GPa | [39] |
| Coefficient of restitution | 0.8 | [39] |
| Coefficient of static friction | 0.15 | [40] |
| Coefficient of rolling friction | 0.05 | [41] |
| Particle surface energy | 1.4 mJ/m$^2$ | (Determined) |

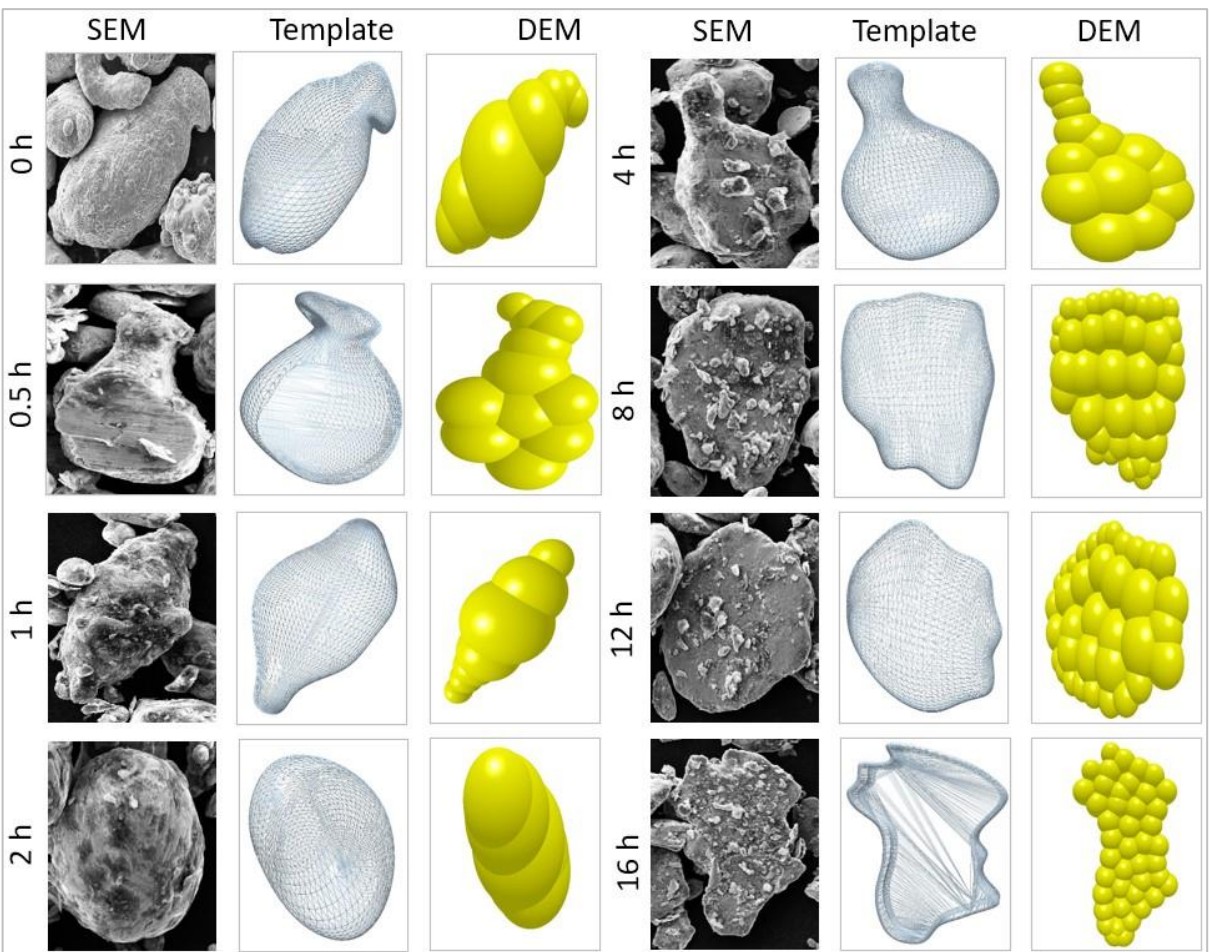

**Figure 4.** Examples of created 3D particles showing the morphological evolution of particle shape from spherical to flat with increasing milling time.

For the experimental measurements, a protective cover for the flowability test kit, an airflow-free room and a silent workplace were selected to minimize the influence of airflow and vibration on the experimental work. The hillside angles were measured, and the mean was taken for each deposited powder, in both the experiments and simulations. Experimental flowability tests were carried out according to the ASTM B213-13 standard. Each test was repeated three times for each powder milling time to achieve a more accurate outcome and conducted with approximately 20 g of powder.

### 2.4. Compaction Characterisation

The apparent and tapped volumes of the samples from as-received powder at each milling time (0.5 to 16 h) were measured to analyze the compaction characteristics of the composites. The tapped volume was obtained by tapping the powder 500 times while placed inside a 50 mL scaled cylinder tube. Measured volumes before tapping (apparent density) and after tapping (tapped density) were used to calculate Carr's Index (*CI*) and the Hausner Ratio (*HR*) using Equations (6) and (7) [42,43]:

$$CI\ (\%) = 100 \times (\rho_t - \rho_a)/\rho_t \tag{6}$$

$$HR = \rho_a/\rho_t \tag{7}$$

where $\rho_a$ and $\rho_t$ are the apparent and tapped densities, respectively.

### 3. Results

#### 3.1. Particle Size Distribution (PSD) of the Composites

The PSD of the milled composites after different milling times (from 0.5 to 16 h) are shown in Figure 5a, including the median particle size and volume density (Figure 5b), which were obtained using the Malvern Mastersizer-3000 machine. While the median particle size progressively increases with milling time, the volume density of the median particle size shows a significant decrease after 4 h milling. The gradual increase of the particle size with longer milling times can also be seen from the shift of the PSD curves to right (see Figure 5a). The rise in particle size can be explained by plastic deformation and flattening of the powder during the mechanical milling process [44]. The volumes of smaller and larger particles also tend toward the median value ($Dv_{50}$), and the volume density increases from 6.3 to 8.1% with an increasing milling time from 0.5 to 4 h. However, beyond 4 h milling, the trend reverses, and $Dv_{50}$ of the 16 h milled powder is reduced to 5.75%. This sharp reduction in median volume density and larger median particles after 4 h milled powder demonstrates that particle welding had initiated in this period. Moreover, the large surface area of the flattened particles also dominantly affects the particle size and volume density. It has been reported that laminar structures might cause fluctuation in the particle measurement owing to the angle between the laminar particle and the laser beam of the analyzer [44]. Additionally, some other studies (for instance, $Y_2O_3$-reinforced AA7075 and Ti-based alloy [45,46]) exhibit a similar fluctuation at the PSD pattern with shifting and variation in volume density due to the flattening of the powder, cold-working and fracture mechanism. Therefore, locally accumulated and disintegrated Gr can also cause this fluctuation because of the surface area of the GNPs. Furthermore, the progressive curve shifting of the $Dv_{50}$ to the right shows that welding mostly occurs in smaller particles, which tend to get larger due to the compressive forces from the milling balls [47].

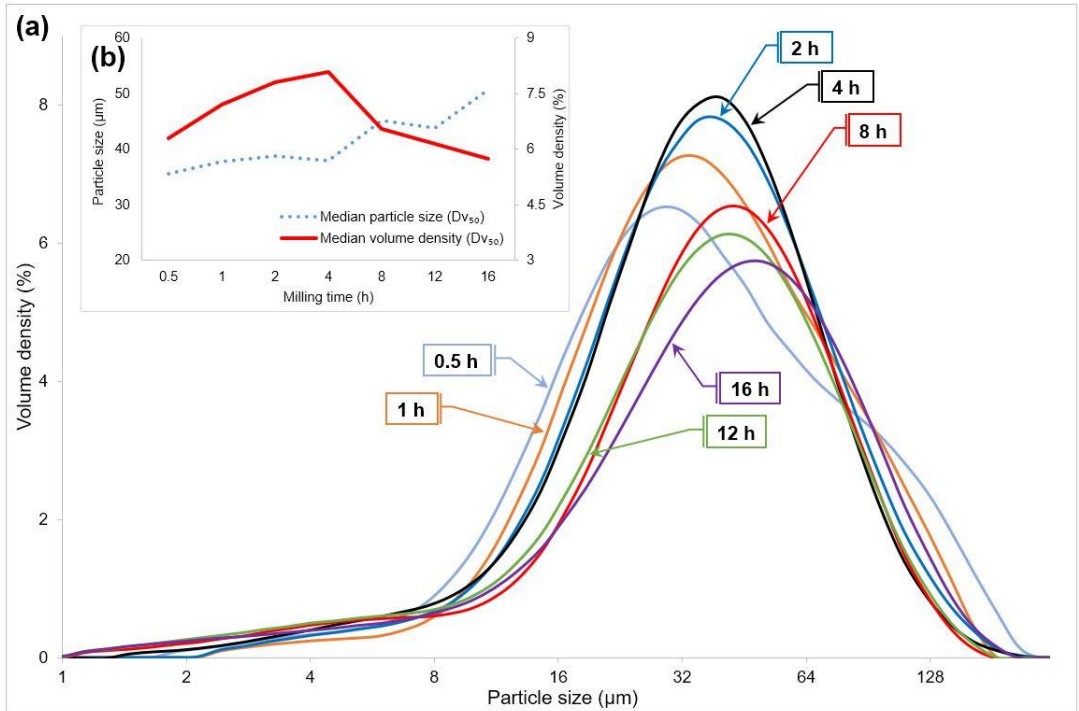

**Figure 5.** The plot of (**a**) PSD and (**b**) median values of milled composites from 0.5 to 16 h.

#### 3.2. Microstructural Characterisation

The XRD pattern of the 0, 4 and 16 h milled composites in Figure 6 shows that the patterns are nearly identical. The "0 h" refers to the average crystallite size and lattice strain of the as-received powder, without milling and GNP reinforcement. The intensity of the

16 h milled powder is contracted and broadened compared to the others. This broadening (which is also related to a reduction in the size of the coherent domain [48]) is caused by the presence of a high dislocation density and other crystalline defects, created during the mechanical milling [49]. Additionally, this contraction substantiates the idea that crystallite density was enhanced with increasing milling time [50]. The average crystallite size of the alloy powder is estimated using Equation (1) as 35.75, 33.93, and 27.14 nm for the 0, 4 and 16 h milled powder, respectively. Five peaks of AA2024 were detected at two-theta of 38.5°, 44.7°, 65.1°, 78.2°, and 82.4°. This is in agreement with previous studies, where grain refinements were observed from the gas atomized as-received powder to the milled powder for AA2124 and AA6005 alloys, because of the mechanical deformation in the milled powder particles [51,52]. No Gr (or carbon) or any other inter-metallic compounds were detected in the XRD test, due to the limitation of the X-ray in identifying phases with a volume fraction of less than 2% [53–55]. Similarly, in some other studies [56,57], no other phases, including alloying elements, in both the as-received and milled AA2024 powder have been observed. The absence of carbon peaks is potentially due to homogeneous dispersion of GNPs within the matrix, amorphization and unfavorable strain/GNPs effect which decreases the peak intensity of GNPs [55,58].

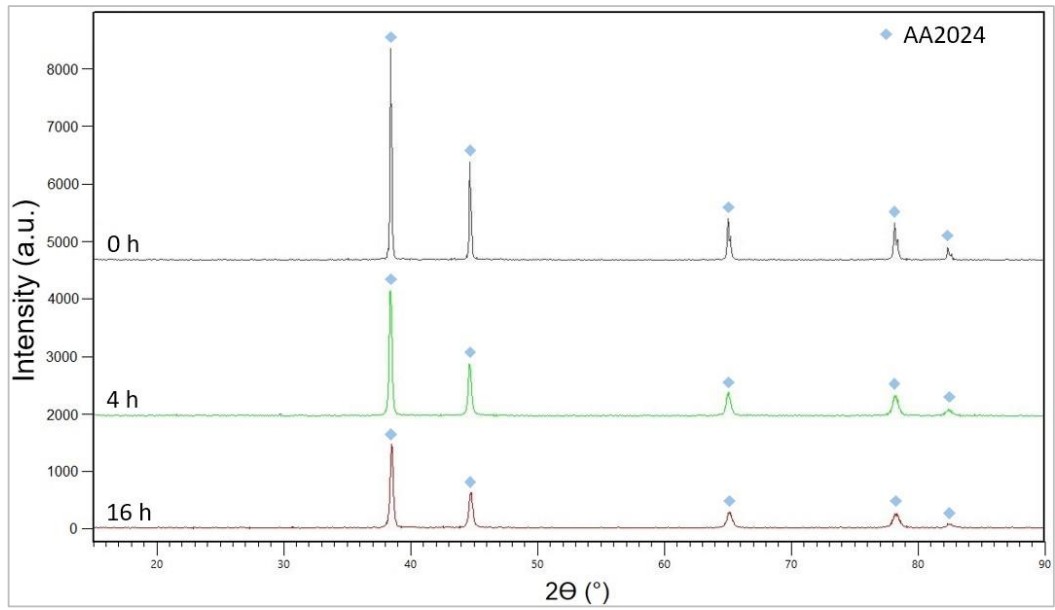

**Figure 6.** XRD patterns of 0, 4 and 16 h milled composites.

SEM images showing the effect of different milling times (from 0.5 to 16 h) on the milled powder under constant milling speed (100 rpm) are shown in Figures 7 and 8. Large agglomerations of GNPs (marked with arrows), apparent at the beginning of the mechanical milling, demonstrate that the distribution of agglomerated Gr flakes is insufficient. Moreover, the Gr flakes did not adhere to the surface of the powders. However, after 4 h milling, the GNPs are initiated to disperse and adhere to the powder surfaces. It is important to note that powder morphology did not change significantly between 0.5 and 4 h milling time, because of the mild impact energy applied under low milling speeds. The other reason for the less plastic deformation of the powder at the beginning stage is the protective role of the PCA. It has been noted that stearic acid creates resistance against cold welding between particles and the accumulation of the particles by coating the particle's surface [59]. Additionally, the particle to milling equipment fracture rate may increase with the addition of PCA, while the friction coefficient between the ball and the powder particle decreases [60]. Therefore, an adequate amount of PCA can prevent the powder from extensive plastic deformation at short milling times [60]. Gr has an intrinsic tendency to form agglomerations due to the large specific surface area, strong Van der Waals attraction and

π-π interaction of Gr [15]. Short milling times promote the formation of agglomerations to begin with, resulting in the impact energy being used to dissolve the Gr particles first. Because of this, shorter milling times tend to result in fewer morphological changes.

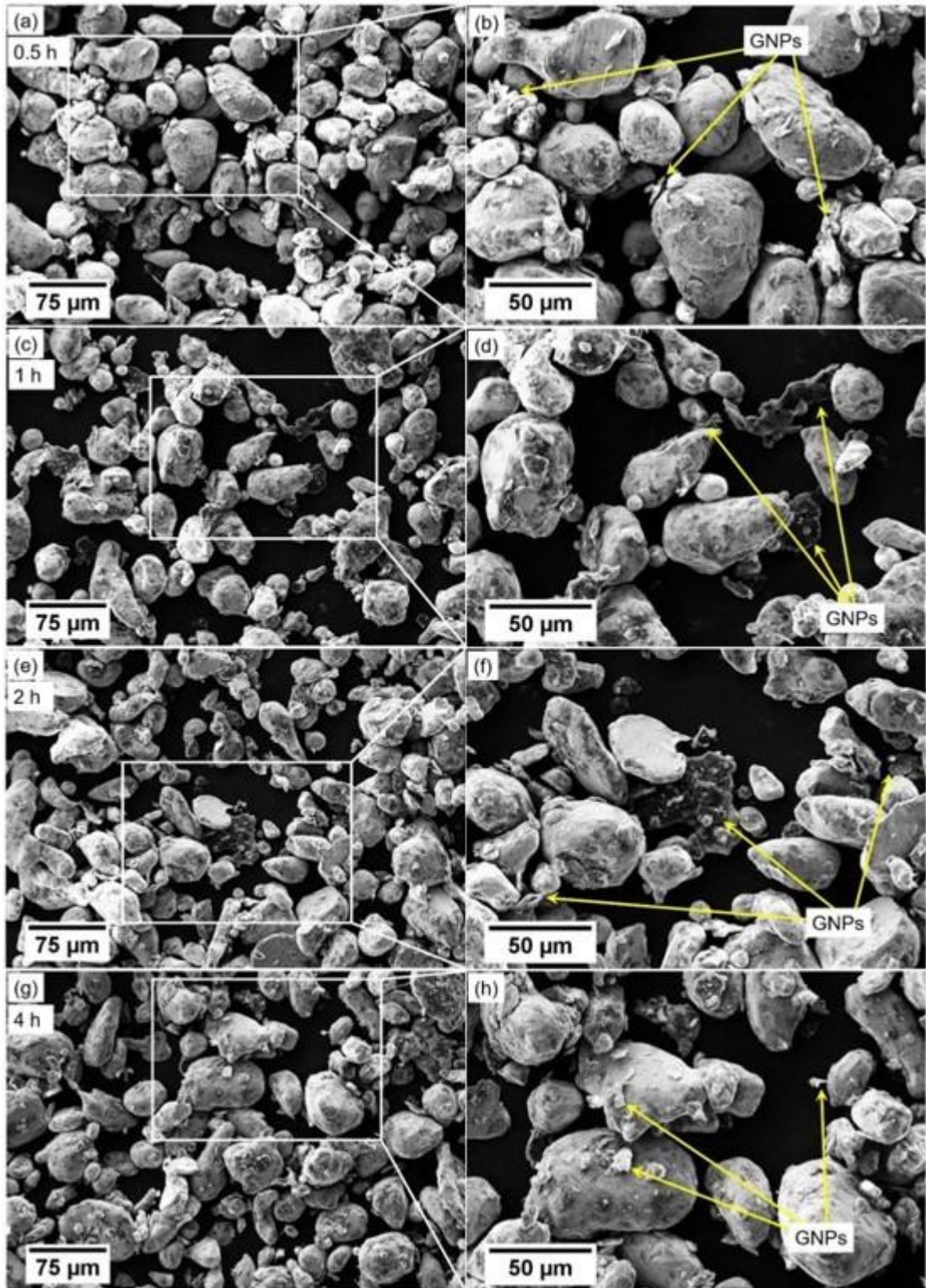

**Figure 7.** SEM images showing the morphological alteration of powders and the dispersion of agglomerated GNPs in the milled alloys for (**a,b**) 0.5 h, (**c,d**) 1 h, (**e,f**) 2 h and (**g,h**) 4 h with magnified images.

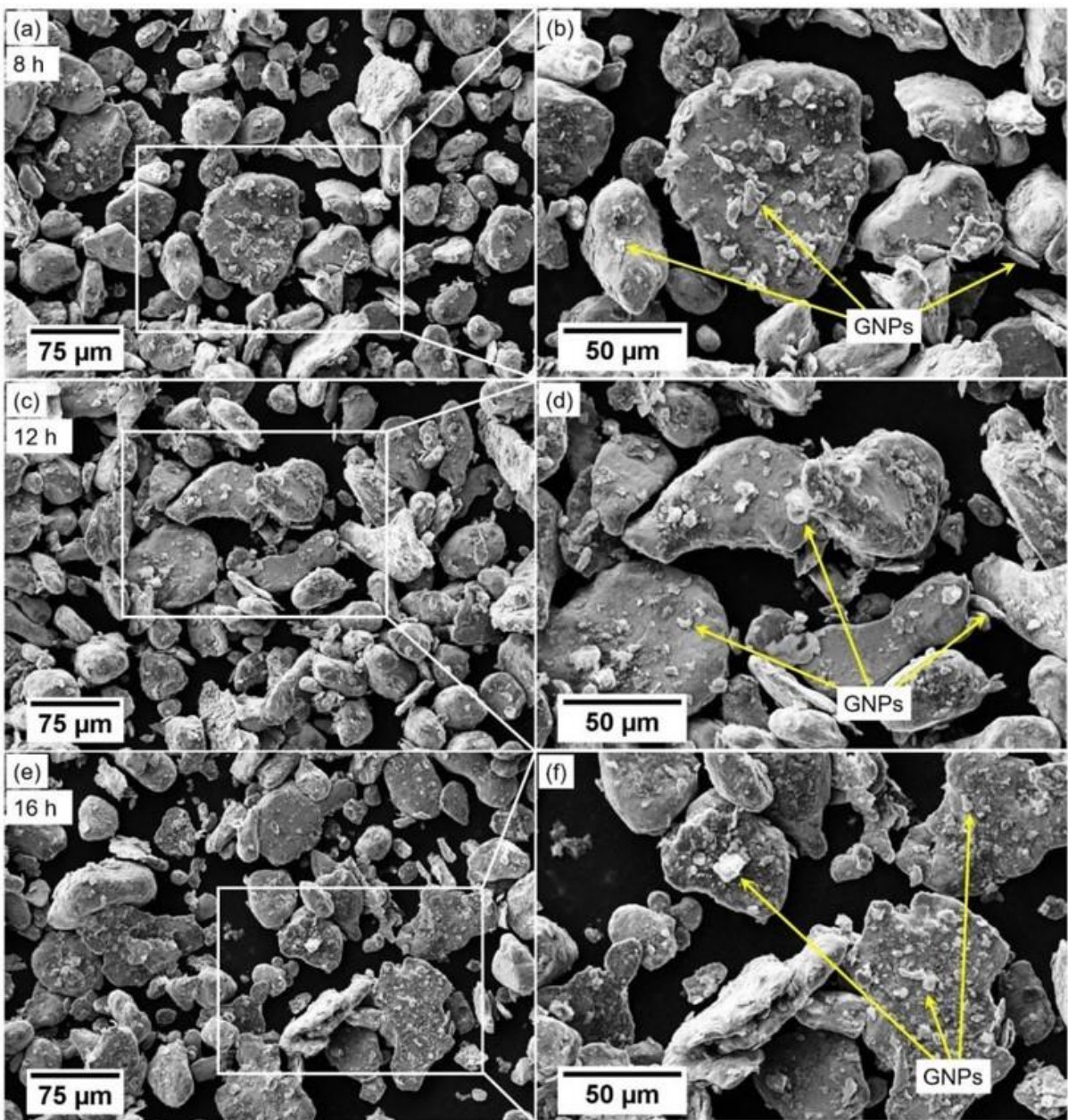

**Figure 8.** SEM images showing the morphological alteration of powders and the dispersion of agglomerated GNPs in the milled alloys for (**a**,**b**) 8 h, (**c**,**d**) 12 h and (**e**,**f**) 16 h with magnified images.

Longer milling times, ranging from 8 to 16 h, visibly changed the powder morphology from nearly spherical to flat, as shown in Figure 8. The plastic deformation of the powder with long milling times becomes more obvious because of the continuously applied impact energy. While the PCA protects the powder against plastic deformation during short milling times (as stated above), the particles cannot tolerate continuous mechanical deformation due to the work-hardening effect at longer milling times [60]. The powder keeps flattening throughout the mechanical milling process, but cold welding did not appear until after 16 h. A similar effect of low-speed ball milling has been observed for CNT- and GNPs-reinforced Al composites milled under low-speed ball milling parameters [61,62]. Flat powder particles negatively affect the powder flowability, which is crucial for AM processes [63]. Extremely irregular and nearly flattened particles cause more voids in both powder bed and deposited powder layer which dramatically increase the porosity of the fabricated parts [64]. Additionally, it has been noted that the large powder size

might provide a rough surface after fabrication due to partially melted powder particles adhering to the part's solidified surfaces [65]. Furthermore, GNP particles were dispersed and adhered to the Al powder surfaces at increased milling times as shown with yellow arrows in Figures 7 and 8. The fine GNPs, homogeneous dispersion, and adherence of Gr flakes to the powder were obtained from 16 h milled composite powder (see Figure 8f).

The average thickness of the raw and milled powders is shown in Figure 9. More than 100 individual powder particles were analyzed using SEM images of the powder at each milling time. ImageJ 1.53e software has been used to measure the particle size and thickness. The thickness of the powder is measured and found to decrease gradually in direct proportion to milling time; however, particle surface area is increasing. The thickness of the Gr particles could not be determined using SEM images because adhered Gr particles lie parallel to the observation surface. In the following sections, the negative effect of flattened particles is analyzed in detail using DEM simulations.

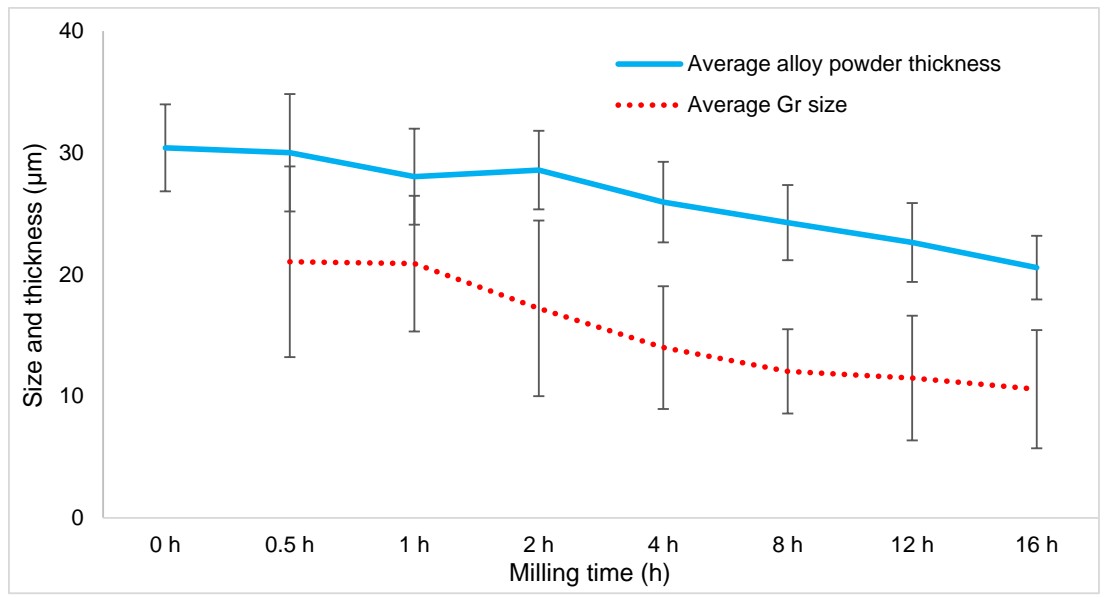

**Figure 9.** Average particle thickness of the as-received and milled composite powder.

### 3.3. Microhardness Analysis of the Powders

Optical microscope (OM) images of the 0, 4 and 16 h milled powders, mounted in resin sample holders are shown in Figure 10. The raw and milled powders were embedded into the resin in order to conduct microhardness tests. The evolution of a flat powder morphology with increasing milling time is visible in the OM images as well as the SEM images. Some powders were exfoliated during the grinding and polishing process, creating small gaps on the observation surfaces.

The Vickers microhardness (HV) test results (under 100 g load and 10 s dwell time) for 0, 4 and 16 h milled powders are given in Figure 11. This test has been conducted on both raw and milled powders in order to demonstrate the evolution of the powders under different milling times. To provide more precise results, the measurements were gathered from different horizontal and vertical locations. The microhardness of the 0 h increased from 98.9 to 101.8 $HV_{0.1}$ at 16 h milling. This small variation illustrates that low ball milling has no significant effect on the microhardness of the milled powder. On the other hand, the progressive rise in the hardness and reduction in crystallite size in the XRD results (Figure 6) suggests that increased dislocation density inside the particles at longer milling times creates a more refined crystallite structure and internal crystallite strain (microhardness) [12,66]. This also substantiates that a large number of crystallite boundaries causes an increase in the internal strain and restricts the movement of the

crystallite proportionally to the milling time [67,68]. Hence, the hardness of the particles starts to improve progressively.

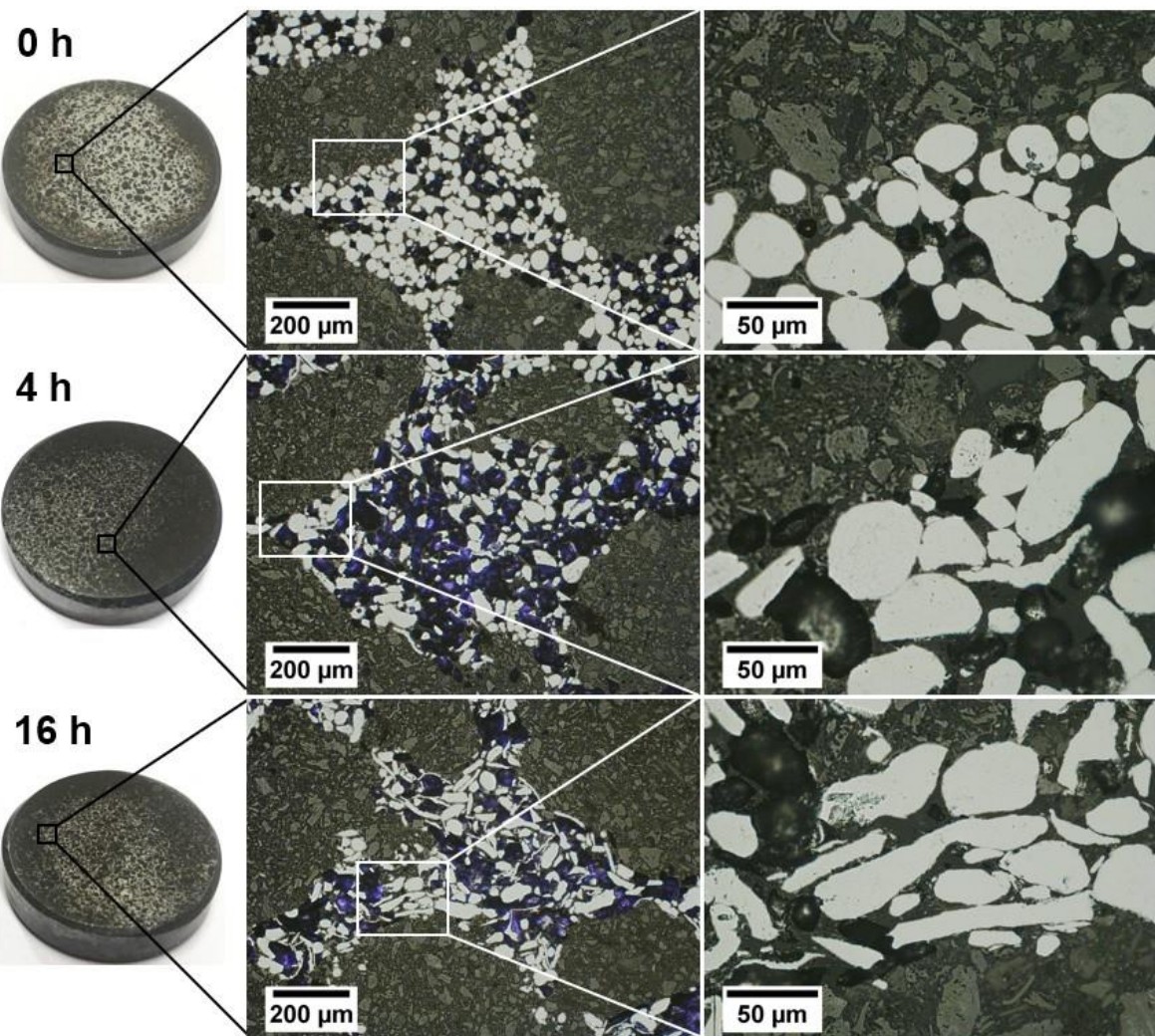

**Figure 10.** OM images of 0, 4 and 16 h milled powders.

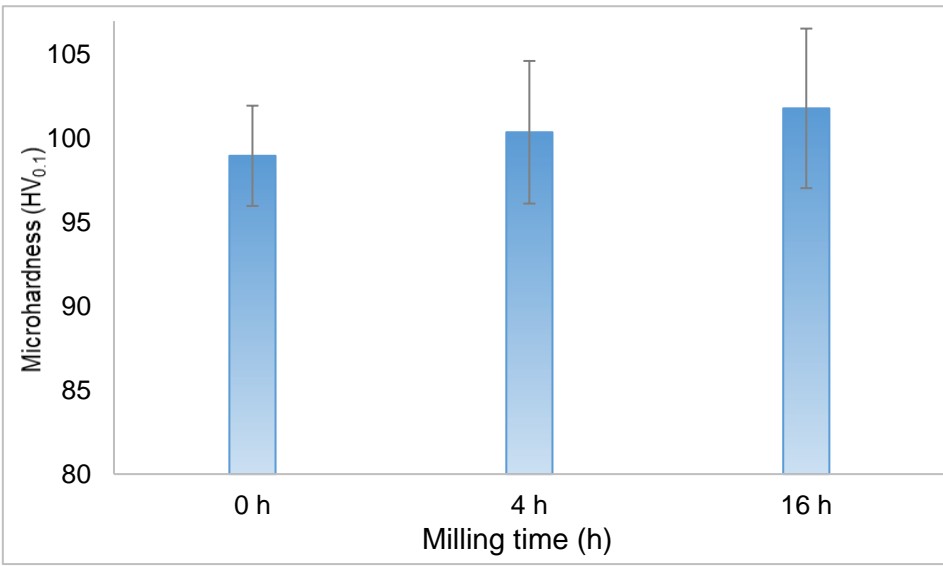

**Figure 11.** Microhardness test results of 0, 4 and 16 h milled powders.

### 3.4. Calibration of the DEM Surface Energy, Particle Type, Volume and Morphology

The calibration process involves four stages. First, the surface energy of the as-received powder is determined using experimental data. Then, commonly used particles (Type 1) and realistic particles (Type 2) are compared in terms of alloy flowability. After, different particle volumes are simulated using DEM. Finally, a wide range of particle morphologies are studded.

Before DEM simulations can be undertaken, it is essential that the model parameters are calibrated to the material under consideration. In the first stage of the calibration process, the DEM simulation was driven with three different as-received powder particle morphologies (obtained from SEM images of real particles) under different surface energy values in order to calibrate the powder-to-powder surface energy (see Figure 12). Most of the AA2024 particles have asymmetrical forms and irregular shapes including oval, fine, nearly spherical, curved, elongated, satellited and others. Particles that were chosen for calibration and flowability analysis have been carefully selected to contain most of these particle types. Increasing the particle surface energy (from 0 to 2 mJ/m$^2$) results in a higher angle of repose (from 7.1° to 36.5°), which means that the flowability of the powder becomes poor at higher surface energy values [69]. The closest hillside angle to the experimental result of the as-received powder (27.4° was achieved with a surface energy of 1.4 mJ/m$^2$ (28.5°, see Figure 12c). A similar approach has been followed in some other studies [29,69] to determine the surface energy from the angle of repose achieved from physical funnel experiments. As-received powder was used to calibrate the surface energy value and volume differentiation. The surface energy of 1.4 mJ/m$^2$ was adopted for the rest of the study.

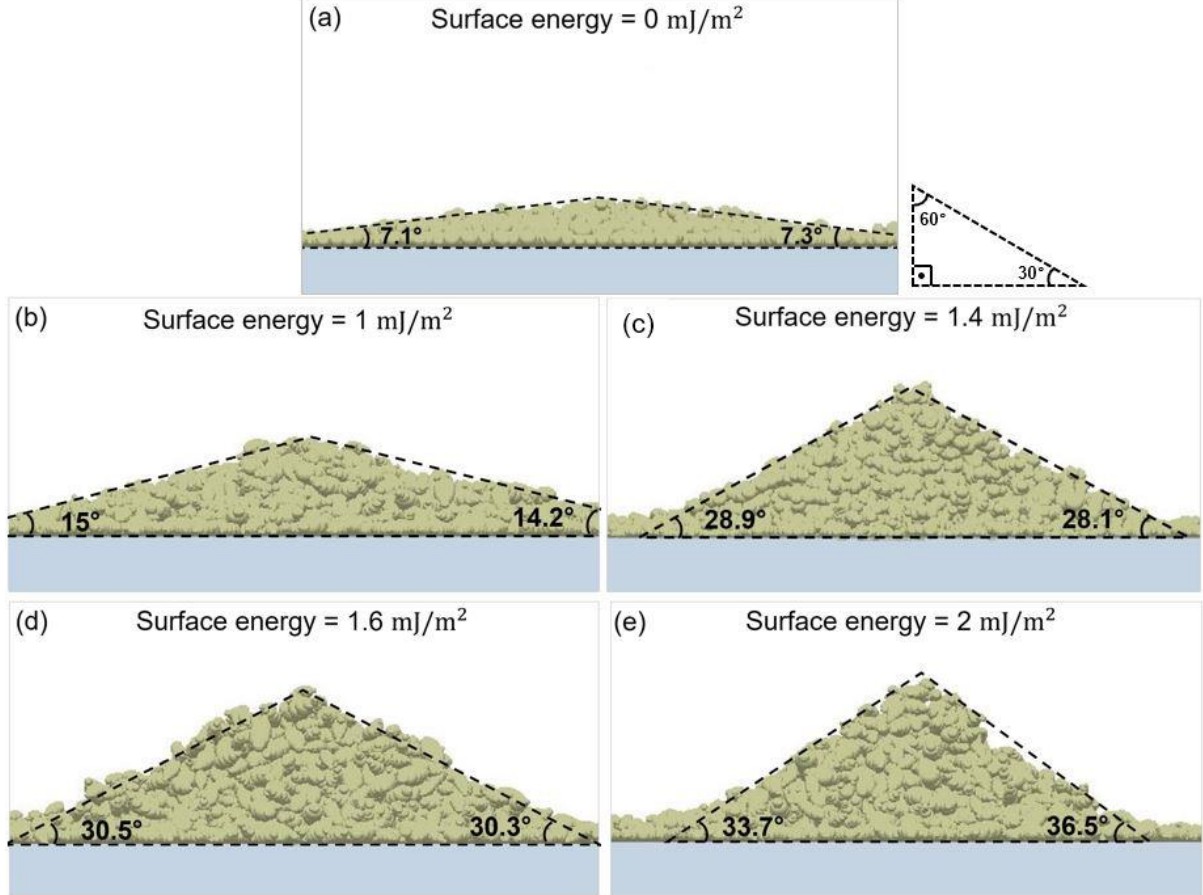

**Figure 12.** The hillside angle of the as-received powder under surface energy values of (**a**) 0, (**b**) 1, (**c**) 1.4, (**d**) 1.6 and (**e**) 2 mJ/m$^2$.

The suitability of the commonly used particles (Type 1) for DEM simulations of AA2024 powder was tested in the next step of the calibration (see Figure 13). First, single spherical and five different multi spherical particles were simulated individually, and then all particles were simulated in a powder pool (Type 1a–f) with equal percentages (16.6%). Even though the hillside angle of the particles is lower than the experimental result, Type 1e has a higher value than experimental work due to the flat powder morphology. Additionally, an equal combination of the series in the simulation powder pool resulted in a higher value than the experimental work. As a consequence of this, none of the Type 1 series particles and combination of the series is considered representative for the DEM simulation of AA2024 alloy. In order to have an accurate result for the alloy, SEM images of the real particles need to be utilized.

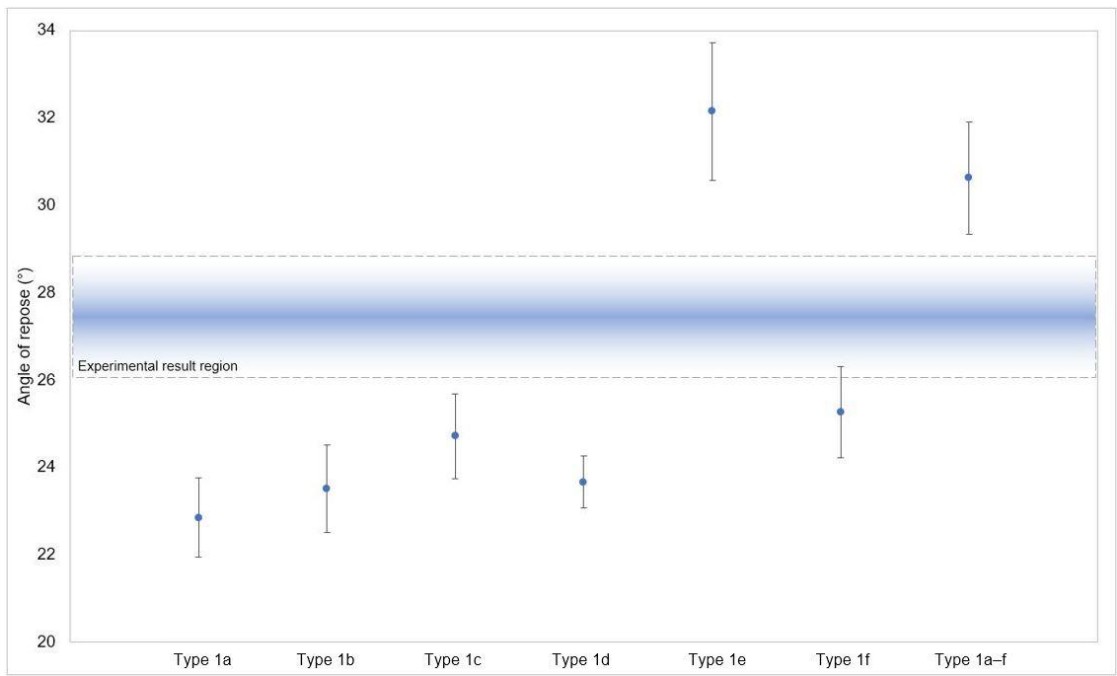

**Figure 13.** Angle of repose of commonly used particles (Type 1) in the literature.

The percentages of three different particle morphologies in the powder pool of the DEM simulation were varied in order to estimate the effect of the volume of different particle morphologies on the angle of repose. The results are shown in Figure 14. Dominant particles in the powder pool exhibit identical patterns, such as the similarities of both Type 2a and Type 2b (50%). Moreover, it shows that a multi-shape particle with varied percentages in the powder pool is not strongly influential on the angle of repose. To this end, an equal volume (33.3%) for each particle in powder pools was used in the rest of the study.

In the last stage of the calibration process, three different particle morphologies (Type 2a–c) were verified with six (Type 2a–f) and ten (Type 2a–j) different particle morphologies (see Figure 15), again based on SEM images. The DEM results using six and ten different particle morphologies are almost identical to the results using three different particles, and all are within the experimental result region. This resemblance suggests that selected three particles are sufficiently representative to simulate the AA2024 powder. Extended particle morphology results in less deviation from experimental results. Similarly, it has been observed that a more detailed representation of real particles in DEM simulations produces findings that are close to the experimental data [70]. On the other hand, while each three-particle simulation takes 2–3 days, a ten-particle simulation takes 7–10 days to complete. Accordingly, when the long simulation time and extensive preparation period for a large variety of powder morphologies in the simulation powder pool were consid-

ered, three-particle morphology was representative enough and gives adequate results in comparison to the six and ten morphologies.

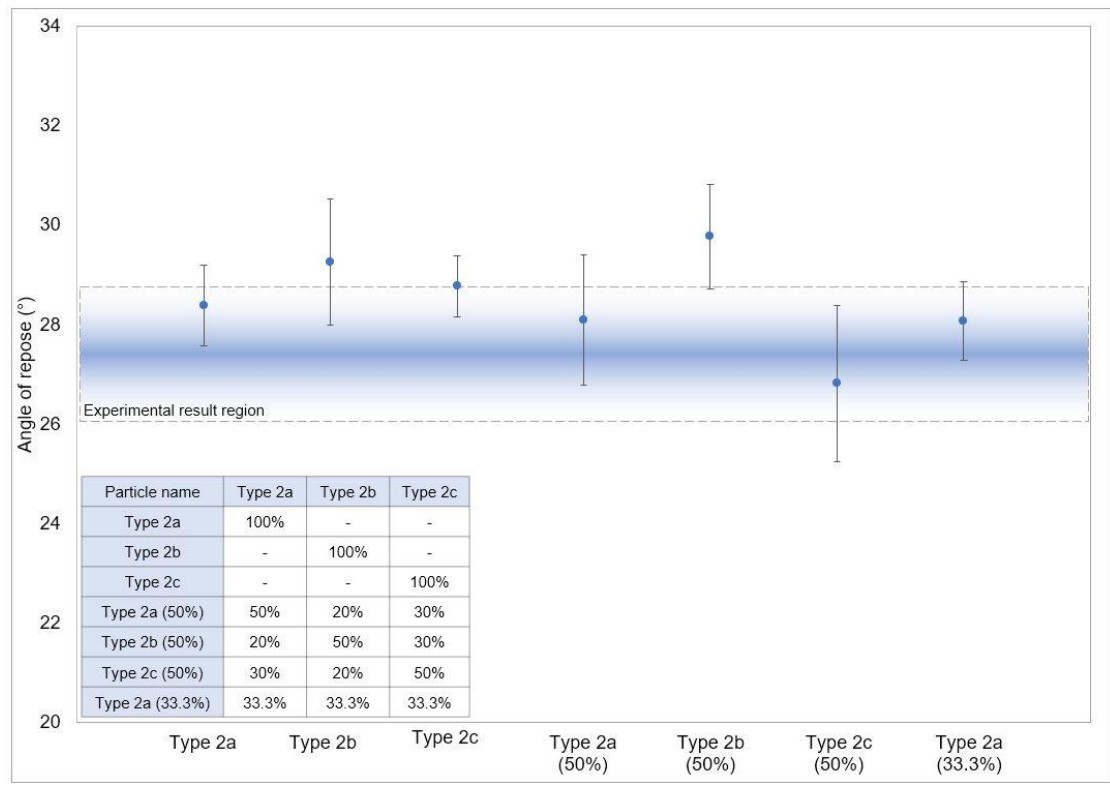

**Figure 14.** Angle of repose for different percentages of powder morphologies (Type 2a–c) in simulation powder pool.

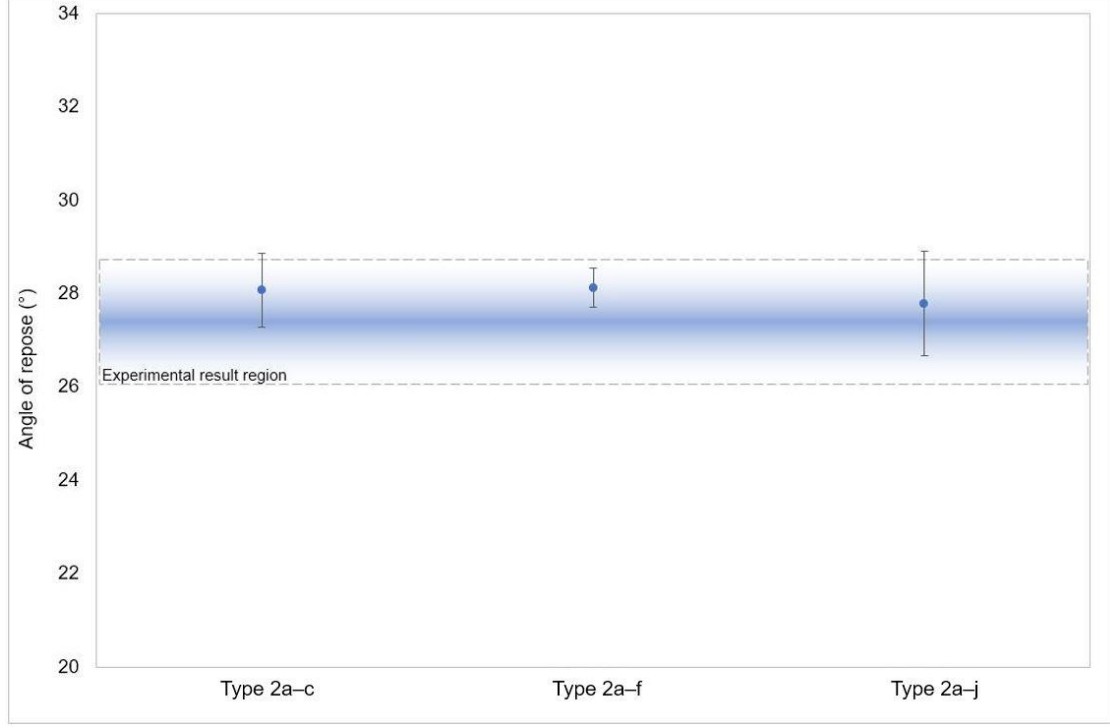

**Figure 15.** Angle of repose of three (Type 2a–c), six (Type 2a–f), and ten (Type 2a–j) powder morphology obtained from SEM images of real particles.

### 3.5. Flowability Analysis

Figure 16 shows the angle of repose of raw and GNPs-reinforced alloy under different milling times (0.5 to 16 h). The narrower hillside angles represent better flowability of the powders [71]. Experimental results show that as-received powder has the narrowest hillside angle (27.4°) compared to the milled powders' angles. Milled alloys range between 31° and 36.4°; thus, milled alloys are less flowable than as-received powder. Initially, the hillside angle of the milled alloys in the experiment slightly reduces from 33.7° to 31.1°, with increasing milling time from 0.5 to 4 h. After 4 h, the angle started to increase up to 35.8°. On the basis of the above observation, it can be deduced that reinforced Gr, which has a poor flowability (see Figure 17), negatively affects the flowability of the as-received powder owing to the agglomerated Gr particles in the powder bed at the beginning. However, the large and accumulated GNPs disintegrate and adhere to the powder surface at further milling, thus becoming less effective on the angle of repose. Additionally, the morphological evolution of the powder from nearly-spherical to nearly-flat could also affect flowability. Similar observations and findings have been achieved and reported with multi-layer Gr-reinforced Ti-based MMC under different milling times [7]. The contrast between the experimental and simulation results (when created from real particle shapes) did not provide a significant difference in flowability. However, the simulation results at longer milling times (in which separated Gr particles have less effect on flowability) are more comparable to the experimental results. The low weight ratio of the Gr (0.5%) in the composite is another reason for the insignificant effect of the Gr. The effect of Gr percentages on the flowability of Gr/Inconel 718 composite has been studied experimentally by [72], which reported that 0.25% and 1% of Gr negatively affect the flowability 2.2% and 9%, respectively.

Meanwhile, the continuously applied impact energy on the powder inside the milling bowls gradually changes the powder morphology to a nearly-flat particle. The SEM images (see Figures 7 and 8) corroborate this. Nevertheless, after 4 h, the angle of repose starts to increase up to 35.8° due to a dramatic change in the particle shape at this point, from nearly spherical to nearly flat, which reduces the flowability characteristic of the powders. Similarly, graphene-oxide (GO) nanosheets have been studied as reinforcement material for Cu, and it was reported that the addition of GO to the matrix material results in an accumulation of GO at a low milling time (1 h) [73]. However, further milling (up to 5 h) dispersed the GO and the smaller particles adhered to the matrix material's surface, which then becomes less effective on the powder flowability [73].

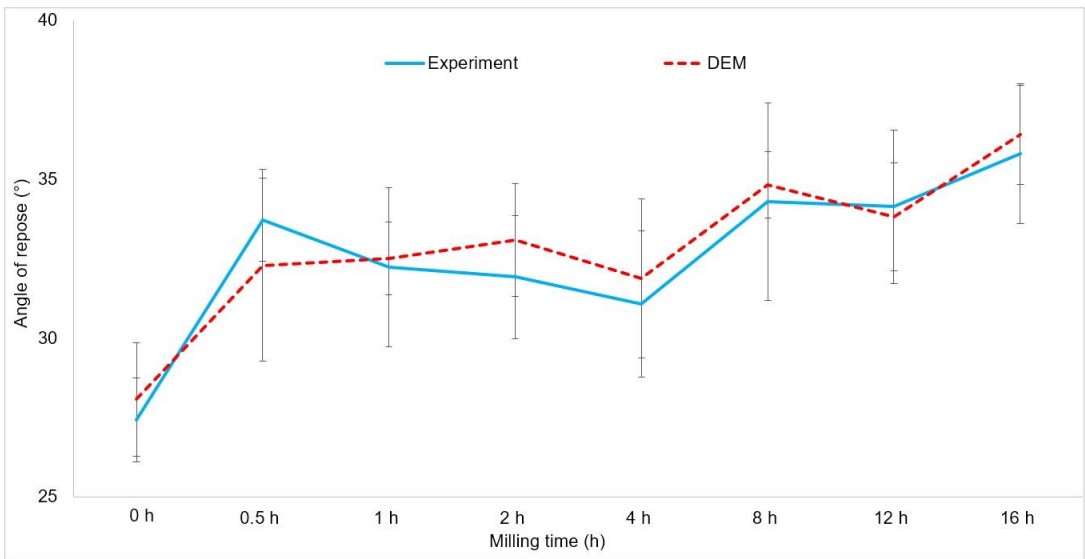

**Figure 16.** Angle of repose of the as-received powder and milled Gr/AA2024 obtained from experimental work and DEM simulation.

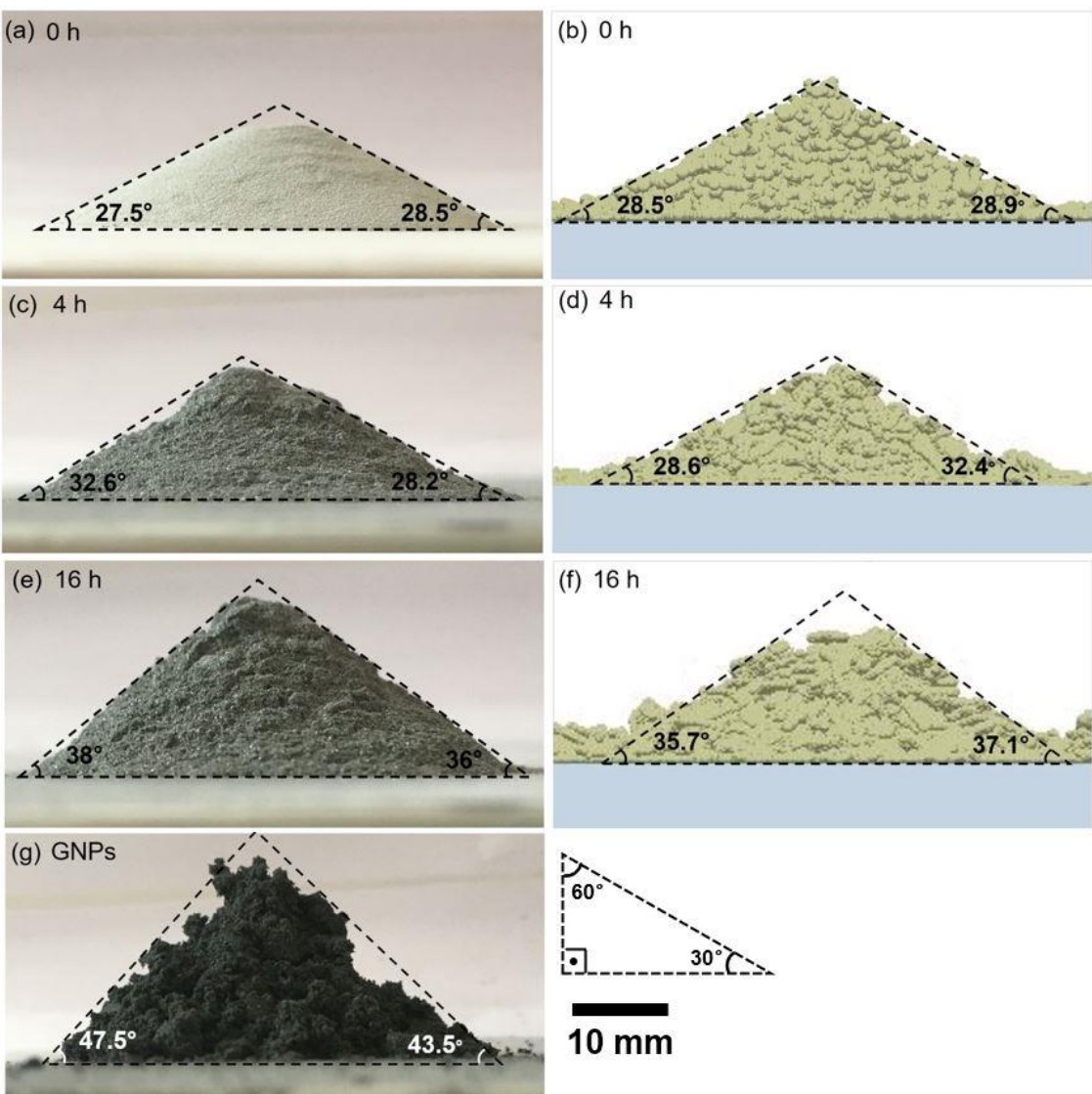

**Figure 17.** Reposed powder, showing GNPs, 0, 4 and 16 h milled alloy from (**a,c,e,g**) experimental work and (**b,d,f**) DEM simulation.

Flowability characteristics, in relation to the hillside angle, Carr's index, and the HR are given in Table 3. The measured hillside angle of the powders indicates that, while as-received powder (27.4°) has excellent flow characteristics, milled powders (ranging from 31° to 36.4°) are in the free-flow characteristic category. Carr's index and the HR again show the excellent-flow characteristic of the as-received powder. However, milled powders lie between free-flow and poor-flow characteristics.

**Table 3.** Flowability characteristics in relation to the hillside angle, Carr's classification, and the HR [27,74].

| Flow Characteristics | Hillside Angle | Carr's Index | Hausner Ratio (HR) |
| --- | --- | --- | --- |
| Excellent-flow | <30° | <10% | 1–1.11 |
| Free-flow | 30–38° | 11–15% | 1.12–1.18 |
| Fair-flow | 38–45° | 16–20% | 1.19–1.25 |
| Poor-flow | 45–55° | 21–25% | 1.26–1.34 |
| Very poor-flow | >55° | >26% | >1.35 |

Some illustrative images of the hillside angle from experimental work (left side of Figure 17) and simulation (right side of Figure 17) are given for comparison. The excellent-flow characteristic (27.4°) of the as-received powder and the poor-flow characteristic (46.3°) of the GNPs are shown in Figure 17a,g. The borderline-excellent flowability of 4 h milled alloy (31°) and borderline-fair flowability of 16 h milled alloy (36°) are represented in Figure 17c–f.

Compaction results for the raw and milled alloys are given in Figure 18. Both Carr's index and the HR of the powders depict that milled powders are more compressible than as-received powder. In the literature, a HR over 1.25 is considered to be a sign of poor flowability [75]. While the flow characteristic of the 0 h is excellent, 1 and 4 h milled powders have free-flow characteristics. However, the compressibility of the 12 and 16 h milled alloys show poor-flow characteristics due to the flat powder shapes. Both the hillside angle curves (see Figure 16) and compaction characteristic curves (see Figure 18) are nearly identical, showing similar development of the powder with milling time.

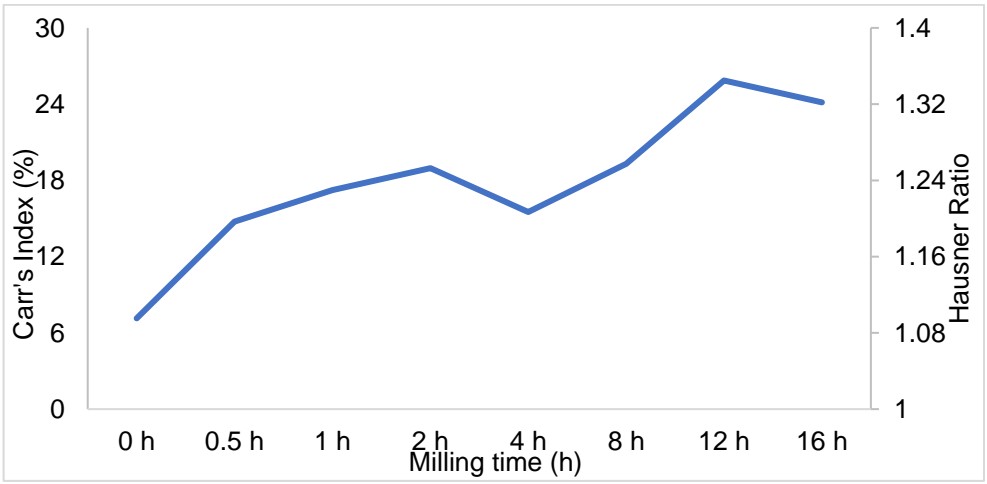

**Figure 18.** Carr's index and the HR of the composites at different milling times.

### 3.6. Effect of Gr Concentration on Flowability

Figure 19 depicts the comparison of the hillside angle with different percentages of Gr reinforcement in the composite powder at 4 h milling. Additionally, the angle estimated from the DEM simulations is shown for comparison. Even though the angle of repose depends on the concentration of Gr in the composite powder, the hillside angles of the composite powders are relatively close to each other. While 0.1% of Gr-reinforced composite has the poorest flowability, 1% of Gr-reinforced composite has the best flowability. This improvement in flowability with more Gr concentration from 0.1 to 1% provides grounds to hypothesize that more Gr in the composite consumes more energy, in order to break the strong bonds between Gr sheets; however, the energy generated during the milling with a low percentage of Gr present (i.e., 0.1%) will be used to change the morphology of the powder to flat. Even though slightly flattened powder, corresponding to the reduction in the Gr percentage, affects the angle of repose of the composite at a low percentage of Gr, the powder remains in the free-flow region. In contrast to 0.2% Gr, there is a 0.4° rise at 0.5% Gr. Possible local accumulations of Gr in the composite might cause some small fluctuations. Despite the established dependence, a further examination of the relationship between higher Gr concentration and flowability of the composite powder is needed to fully understand it.

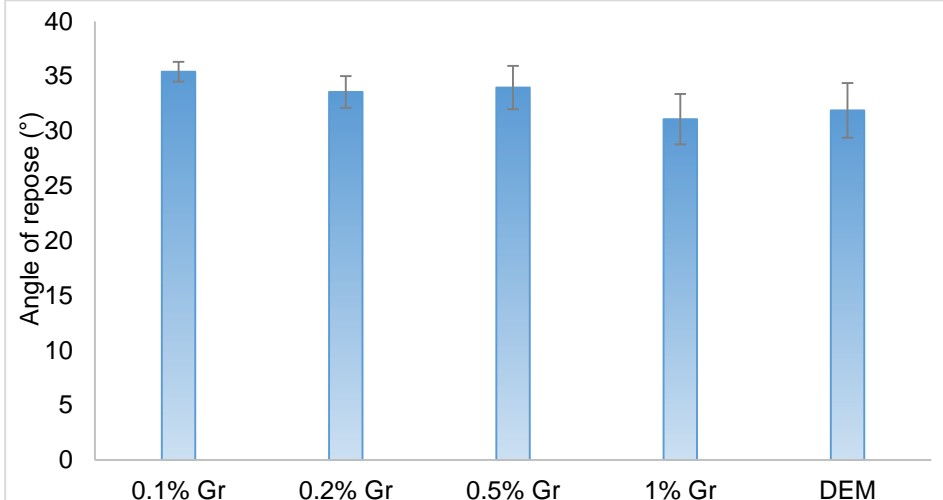

**Figure 19.** The hillside angles of 4 h milled powders with different percentages of Gr (0.1, 0.2, 0.5, and 1 wt.%) and DEM.

## 4. Discussion

The PSD of the milled powders (Figure 5) demonstrates that welding between powder particles is seen in smaller particles because of the compressive force of the milling equipment, especially after 4 h of milling time [47]. Hence, the curves gradually shift to the right. A similar effect has been observed for Al-nitride and Si-nitride reinforced AA6061 [44]. The powder morphology also significantly affects the median particle size and volume density, and accumulated Gr particles at short milling times under LEBM create a laminar structure, because there is insufficient compressive energy to break the interlayer Van der Waals forces of GNPs. It also has been noted that powder particles smaller than 100 μm are prone to adhering to each other, creating bigger particles, because the Van der Waals forces are predominant as the major contacting forces [71]. Further milling breaks this force, allowing the particles to be separated into smaller particles again. This laminar structure can cause fluctuation in powder size and volume density due to the angle between the particle and laser beam of the analyzer [44]. However, when the impact energy is continuously applied to the powder, GNPs disintegrate into small particles (the laminar structure dissolves in the powder pool) and gradually cover the surface of alloy powder. The shear force created by the collision of milling balls [13,61] has the potential to exfoliate Gr layers from graphite, as well as change the morphology of the milled powders into flake-like morphology (which can also give additional dispersion for GNPs) [76,77]. The micro-forge mechanism generated by milling balls is responsible for the increasing flattening of the milled powders [62]. As a result, the distribution of Gr became increasingly uniform on the milled powder. Powders coated with disintegrated Gr flakes can be seen in Figure 8. On the other hand, milling for more than 8 h results in a flat powder morphology, which negatively affects the flowability of the powder. The gradual morphological alteration of the milled powder and Gr particles with time can be seen in Figures 7 and 8, showing that powder morphology changes from nearly-spherical to nearly-flat, and that agglomerated Gr particles dispersed progressively.

As a result of longer milling time, 20% more new crystallites are formed in the milled powder, which accelerates the grain refinement process. Hence, the microhardness of the milled powder is improved by 3%. Higher milling time results with strain hardening of the ductile powders owing to more plastic deformation on powder. Similarly, the work-hardening of powders owing to continuously applied impact energy at longer milling times also causes a microstructural evolution [78]. The microhardness test of the milled powders gives identical results to the microhardness of as-fabricated AA2024 samples in another study [79]. The inverse correlation of microhardness and crystallite size can be seen in Figure 20. Similar inverse correlations are remarked by others, as well [12,51,52,66,80].

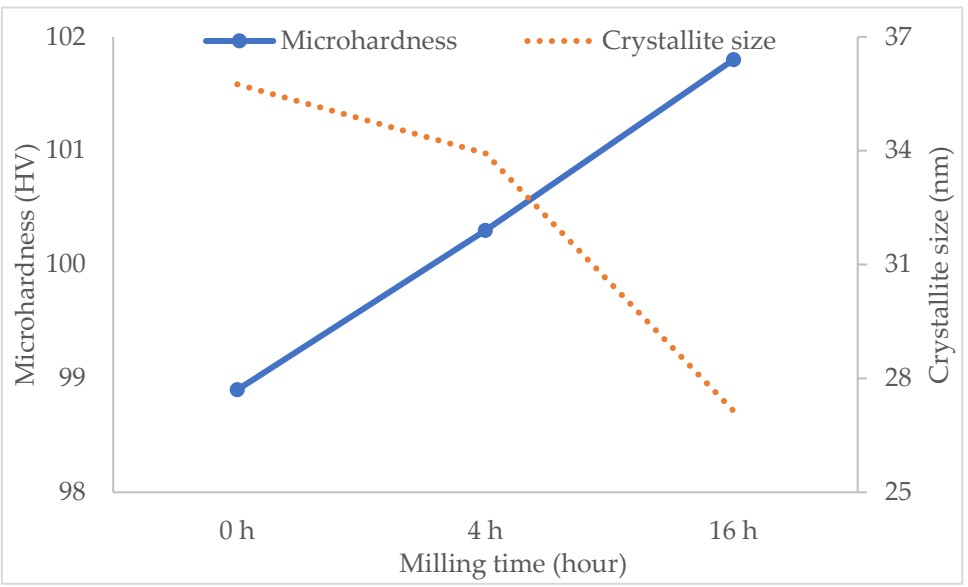

**Figure 20.** Inverse correlation of microhardness and crystallite size via increasing milling time.

AA2024 powders have a unique and non-spherical powder morphology. Type 1 (3) particle morphologies prove that commonly used particles in DEM simulations are not representative of the alloy powder (Figure 13). In order to obtain more accurate results, several powders from SEM images of the real powder (Type 2) need to be used. In the literature, a similar method has been commonly employed for non-spherical particles [81–84]. It is perceptible from Figure 15 that more variety in powder morphology based on SEM images provides more accurate findings; however, longer simulation running time and preparation period of the model with more powders need to be considered. It is also noted that while this method is applicable with powders with similar morphology to AA2024, completely different particle morphology requires new DEM particle modelling owing to the fact that flowability analysis highly depends on powder morphology.

On the other hand, the contrast of experimental study with Gr reinforcement and DEM simulation without Gr gives an opportunity to understand the effect of Gr on flowability. The DEM simulation was calibrated with as-received alloy without milling and Gr reinforcement. Further experimental contrasting discloses the effect of Gr in the composite. This contrast demonstrates that short milling times have more fluctuation than long milling times in comparison to experimental results. The hypothesis behind this fluctuation is that accumulated large Gr particles are able to change flowability under 2 h of mechanical milling. It is a well-known fact, and also proven in Figure 16, that while near-spherical particles give better flowability, near-flat morphology reduces the flowability of the powder [85]. The accumulated and folded Gr particles in the composite at short milling times reduce the flowability of the composite because Gr acts as a flat particle in the composite. However, when the Gr particles were disintegrated into small flakes and adhered to the composite powder, the direct effect of the Gr in the composite was gradually reduced and even disappeared. In the meantime, flattened powder at longer millings deteriorated the flowability of the milled powder. The contrasting experiment and simulation can demonstrate that this modelling approach can be applied for GNPs-reinforced MMCs under particular conditions in order to predict the flowability for further studies.

While the Gr concentration in the composite powder is high, the flowability of 4 h milled powder decreases (Figure 19). The hypothesis behind this reduction is that the energy created inside the milling bowls was firstly used to break strong Van der Waals forces between layers and disperse the accumulated large amount of Gr particles present when 1% Gr is included in the composite prior to morphological evaluation of the powder. However, a small amount of Gr (0.1%) in the composite required less energy to disperse the Gr into the composite. The remaining energy for the 0.1% Gr-reinforced composite

was therefore used to deform the powder morphology of the matrix material. Therewith, more flattened powder negatively affects the flowability of the powder. Further studies are needed to explore the effect of reinforcement element on MMC. Nevertheless, the overall effect of Gr content on flowability is reasonably small, when varied between 0.1 to 1%. On the other hand, 30° is the border between excellent-flow (<30°) and free-flow (30–38°) characteristics. Some measured angles at short milling times (up to 4 h), therefore, tend to cross between flow categories because their angle of repose is close to the border. Another reason for crossing between flow categories is long error bars owing to the variation in the experimental results.

## 5. Conclusions

This study has investigated the effect of milling time on GNPs-reinforced AA2024 powder for use in LPBF. Moreover, agglomeration of the Gr in MMCs with different milling times has been studied through experimental and modelling studies. The following findings are drawn from experimental and simulation results:

1. Short milling times (below 4 h) provided insufficient impact energy inside the milling bowls to separate the agglomerated Gr particles. However, long milling times (over 4 h) notably changed the powder morphology from nearly rounded to flat. Furthermore, GNPs were better separated and adhered to the Al powder surface at longer milling times.

2. For 16 h of milling, there was a slight increase in microhardness (3%), as well as a reduction in average crystallite size (24%) in the milled powder (see Figure 20). This inverse correlation suggests that grain refinement gives additional strength to the milled powder due to the improved internal crystallite strain.

3. Common DEM particles (Type 1) in the literature and near-shape SEM particles (Type 2) were tested and it was found that Type 2 particles are more representative and provide accurate results than Type 1. Additionally, more variety in morphology results in sensitive results; however, the running time of the simulation and preparation of more particles is extensive. It was concluded that experimental work for flowability tests of irregular and non-uniform particle morphologies is faster and more reliable than simulation.

4. The contrast of the experimental work and simulation results regarding the flowability test show excellent correlation at long milling times (Figure 16). However, simulation results at short milling times show differentiation from the experimental results. The reason behind this is that separated Gr particles at long milling times become less effective on the angle of repose. These findings lead us to conclude that the flowability of the composite more depends on powder morphology than the existence of Gr at longer millings.

While better dispersion of Gr is achieved between 4 and 16 h, less morphological alteration is obtained between 0.5 to 4 h. Based on dispersion, morphology, simulation, and experimental results, 4 h of milling under 100 rpm milling speed satisfies the LPBF process requirements to obtain better-deposited layers and uniform composite. Future work will be undertaken using 4 h milled powder under different weight ratios of Gr as a reinforcement material. The mechanical properties and microstructure of the as-fabricated samples will be analyzed in order to find the optimum percentage of the Gr in the MMCs.

**Author Contributions:** M.A.P.: Conceptualization, original draft preparation, review and editing, visualization, methodology, investigation, software, formal analysis. R.S.: conceptualization, review and editing, methodology, supervision, formal analysis, project administration, data curation. M.R.: review and editing, methodology, supervision, data curation. H.G.: review and editing, data curation. Q.H.: review and editing, methodology, data curation. D.G.: conceptualization, review and editing, methodology. All authors have read and agreed to the published version of the manuscript.

**Funding:** This research is supported by the Ministry of National Education of Turkey and ASTUTE 2020 (Advanced Sustainable Manufacturing Technologies). This operation, supporting manufacturing companies across Wales, has been part-funded by the European Regional Development Fund through the Welsh Government and the participating Higher Education Institutions.

**Data Availability Statement:** Not applicable.

**Conflicts of Interest:** The authors declare no conflict of interest.

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
