# Peer review of "Al-Cu-Mg Alloy Powder Reinforced with Graphene Nanoplatelets: Morphology, Flowability and Discrete Element Simulation"

_jmmp, doi:10.3390/jmmp6060148_

Round 1
Reviewer 1 Report
Well, there are some questions and problems arose that requires correction and /or explanations:
1. The particle size of the raw materials, GNPs, is 15 μm in this paper. Low magnification image should be given in Figure 1 for explanation
2. How to obtain the medium particle size in Figure 5? The values in the inset seem not to correspond to the curve.
3. Mapping in SEM is applicable to the analysis of element distribution on the plane. The surface of the powders in Figure 7 is not flat, so it is inappropriate to use surface scanning. In addition, how to remove the stearic acid? How to determine that the C elements detected in SEM is GNPs instead of stearic acid?
4. What is the effect of milling time on the thickness of graphite flakes? It should be given in Figure 10.
5. In Figure 20, how to explain the abnormal increase of the Angle of reply in 0.5% GNPs?
6. In previous studies, it was usually hoped to reduce the thickness of GNPs through ball milling in order to obtain graphene. What is the purpose of low-energy ball milling in this study?
7. All figure references in this article are incorrect, please correct them carefully.
Reviewer 2 Report
1. It is noted that your manuscript needs careful editing by someone with expertise in technical English editing paying particular attention to English grammar, spelling, and sentence structure so that the goals and results of the study are clear to the reviewers.
2. Line 53, “Ball milling offers a wide range of parameters”. why is that?
3. Line 83, “In order to explore these challenges”. What are these challenges?
4. Line 414, “In the first stage of the calibration process, the DEM simulation was driven with three different as-received powder particle morphologies (obtained from SEM images of real particles) under different surface energy values in order to calibrate the powder-to-powder surface energy (see Error! Reference source not found.)”. Are the three powders completely representative of the results for all powder types?
5. Line 426, “Intensive work in order to determine the surface energy has been done with three particles because more particles result in a longer simulation time. While each three-particle simulation takes 2-3 days, a ten-particle simulation takes 7-10 days to complete”. The number of experimental particles was very poorly chosen, and I do not consider the results of this experiment to be reliable.
6. Line 403, “It is important to note the progressive rise in the hardness and reduction in crystallite size in the XRD results”. The present XRD results of the manuscript cannot support the hardness results.
7.Line 662, “For 16 h of milling, there was a slight increase in microhardness (3%), as well as a reduction in average crystallite size (24%) in the milled powder. This inverse correlation suggests that the grain refinement causes additional strength to the milled powder due to the improved internal crystallite strain”. There is no inverse relationship between the grain refinement and the hardness increase.
Reviewer 3 Report
This work is interesting and contains scientific results being of interest for publishing in this journal. However it needs to be substantially improved.
1. It is stated in the line 323 that there should be titanium in the composition of the 2024 alloy which is not detected by chemical analysis of the alloy. But titanium is not indicated as an alloying element of your alloy (lines 113-114). Also the 2024 alloy usually contains Zn and Cr which are also not indicated in your alloy. Please, specify the composition of the studied alloy.
2. What source is mentioned in the line 215? Is it given in References?
3. You show the sizes and shapes of aluminium powders in detail but for Graphene Nanoplatelets you give only 15 μm particle size. If the particles have irregular shapes, so what kind of size do you show? The length, width or thickness of the particle? What range of the particles sizes is? If it is difficult to measure the size of a separate Graphene Nanoplatelet, provide the size for Graphene Nanoplatelets agglomerates.
4. Why do you give the data on oxygen and carbon distribution in composite powder only after 4 h blending? Did you observe an increase in the content of these impurities after 8-16 h blending?
5. How did you measure the thickness of particles in Fig. 10?
6. In section 3.6 ‘Effect of Gr concentration on flowability’ you describe the experiments with the use of various Gr content, however in sections ‘Introduction’ and ‘Materials and Methods’ you do not set such a problem for investigation. You need to add this information.
The reason for improving flowability of the composite powder with an increase of graphene content is not quite convincing. Perhaps, it is better to conclude that the stated dependence needs to be further studied in detail.
7. Sections ‘Discussion’ and ‘Conclusions’ are very long and contain a lot of repeated conclusions. They can be written more compactly. This is a recommendation, not the remark.
8. It follows from your article that experimental determination of flowability of composite powders takes less time and is simpler than the method of Discrete Element Simulation and not always coincides with the results of this method. Can we conclude that at the obtainment of composite powders with irregular and non-uniform particles shape experimental determination of flowability should be only used, and the method of Discrete Element Simulation is of interest to only separate scientific experiments? Because it is necessary to select again the Simulation parameters for every new situation (powder with any other shape, powder of other alloy + additive, powder of other alloy + additive + additive, etc.), and experimental determination needs the same equipment and calculation procedure.

Round 2
Reviewer 2 Report
All the issues in the first round have been well addressed.
Author Response
Thank you for your feedback.
Reviewer 3 Report
Authors replied to all my remarks, and I recommend this article to be published after some correction:
You need to specify if the chemical composition was provided by the powder producer or it was detected by you and how.
Author Response
Thank you for your feedback.
